# Efficacy assessment of a novel endolysin PlyAZ3a$^T$ for the treatment of ceftriaxone-resistant pneumococcal meningitis in an infant rat model

**Luca G. Valente**[1,2,3], **Ngoc Dung Le**[1,3], **Melissa Pitton**[2,3], **Gabriele Chiffi**[1,3], **Denis Grandgirard**[1], **Stephan M. Jakob**[2], **David R. Cameron**[2], **Grégory Resch**[4], **Yok-Ai Que**[2], **Stephen L. Leib**[1] *

1 Institute for Infectious Diseases, University of Bern, Bern, Switzerland, 2 Department of Intensive Care Medicine, Inselspital, Bern University Hospital, University of Bern, Bern, Switzerland, 3 Graduate School for Cellular and Biomedical Sciences, University of Bern, Bern, Switzerland, 4 Centre for Research and Innovation in Clinical Pharmaceutical Sciences, Lausanne University Hospital, Lausanne, Switzerland

* stephen.leib@ifik.unibe.ch

**Data Availability Statement:** All relevant data are within the paper and its Supporting information files.

## Abstract

### Background

Treatment failure in pneumococcal meningitis due to antibiotic resistance is an increasing clinical challenge and alternatives to antibiotics warrant investigation. Phage-derived endolysins efficiently kill gram-positive bacteria including multi-drug resistant strains, making them attractive therapeutic candidates. The current study assessed the therapeutic potential of the novel endolysin PlyAZ3a$^T$ in an infant rat model of ceftriaxone-resistant pneumococcal meningitis.

### Methods

Efficacy of PlyAZ3a$^T$ was assessed in a randomized, blinded and controlled experimental study in infant Wistar rats. Meningitis was induced by intracisternal infection with $5 \times 10^7$ CFU/ml of a ceftriaxone-resistant clinical strain of *S. pneumoniae*, serotype 19A. Seventeen hours post infection (hpi), animals were randomized into 3 treatment groups and received either (i) placebo (phosphate buffered saline [PBS], n = 8), (ii) 50 mg/kg vancomycin (n = 10) or (iii) 400 mg/kg PlyAZ3a$^T$ (n = 8) via intraperitoneal injection. Treatments were repeated after 12 h. Survival at 42 hpi was the primary outcome; bacterial loads in cerebrospinal fluid (CSF) and blood were secondary outcomes. Additionally, pharmacokinetics of PlyAZ3a$^T$ in serum and CSF was assessed.

### Results

PlyAZ3a$^T$ did not improve survival compared to PBS, while survival for vancomycin treated animals was 70% which is a significant improvement when compared to PBS or PlyAZ3a$^T$ (p<0.05 each). PlyAZ3a$^T$ was not able to control the infection, reflected by the inability to reduce bacterial loads in the CSF, whereas Vancomycin sterilized the CSF and within 25 h.

**Funding:** This research was funded by the Swiss National Science Foundation, Grant numbers 176216 (to Y.A.Q. and G.R.), 166124 (to Y.A.Q.) and 189136 (to S. L. L.). The funders had no role in the design of the study; in the collection, analyses, or interpretation of data; in the writing of the manuscript, or in the decision to publish the results.

**Competing interests:** The authors have declared that no competing interests exist.

Pharmacokinetic studies indicated that PlyAZ3a[T] did not cross the blood brain barrier (BBB). In support, PlyAZ3a[T] showed a peak concentration of 785 μg/ml in serum 2 h after intraperitoneal injection but could not be detected in CSF.

## Conclusion

In experimental pneumococcal meningitis, PlyAZ3a[T] failed to cure the infection due to an inability to reach the CSF. Optimization of the galenic formulation e.g. using liposomes might enable crossing of the BBB and improve treatment efficacy.

## Introduction

The emergence and the spread of antimicrobial resistance in the last decades are major threats to human health. Alternative treatment strategies targeting antibiotic-resistant bacteria are urgently needed. One such approach is the use of endolysins (or lysins). Lysins are bacterio-phage-derived enzymes with peptidoglycan hydrolase activity and are thereby able to degrade the bacterial cell wall. Recent *in vitro* and *in vivo* results suggest that endolysins might be promising antimicrobial agents. First, they rapidly lyse bacterial cells including those that are multi-drug resistant [1–6]. Second, they act very specifically, killing only strains from closely related species and do not affect off-target commensal bacteria, which makes them less likely to disturb the physiologic microflora [1, 4, 7]. Third, the frequency of resistance emergence seems to be low [8, 9]. Finally, they appear safe; in 2 recent clinical trials, no detrimental effects were observed after topical [10] or intravenous [11] application in humans.

*Streptococcus pneumoniae* is an opportunistic human pathogen that can cause diseases like sinusitis, otitis media, pneumonia, septicemia, and meningitis. Meningitis due to pneumococci is associated with important morbidity and mortality. Despite the availability of appropriate antibiotic treatment, pneumococcal meningitis (PM) is associated with a high case-fatality rate of up to 37% in high-income countries [12, 13]. By the end of the 20[th] century, a concerning increase in antimicrobial resistance was recognized for *S. pneumoniae* [14]. This was mitigated by the implementation of the pneumococcal conjugate vaccines (PCV), which led to a con-comitant decrease in the rate of invasive pneumococcal diseases (IPD) [15, 16] and overall antimicrobial resistance [17–19]. Yet, as high as 30% of infections caused by *S. pneumoniae* remain resistant to at least one antibiotic [20], and cephalosporin-resistance in PM represents an emerging threat [21]. Additionally, serotype redistribution caused by vaccine selection pres-sure has been observed [22–24], leading to the risk of the re-emergence and spread of non-vac-cine serotypes that are resistant to antibiotics.

The efficacy of different pneumococcal endolysins (e.g. Cpl-1 and Pal) has been previously assessed in experimental *in vivo* models for pneumonia [25], endocarditis [26], bacteremia [4, 6], and otitis media [27]. In an animal model of PM, Cpl-1 reduced bacterial loads within the central nervous system (CNS) after a single intracisternal or intraperitoneal injection [28]. Overall, in principle endolysins seem attractive therapeutic candidates to treat pneumococcal infections.

So far, however, the translation of promising preclinical applications of endolysin into clini-cal trials has been strikingly slow. We adapted a well-established infant rat model of PM and evaluated the efficacy of a novel endolysin called PlyAZ3a[T] for the treatment of a ceftriaxone-resistant *S. pneumoniae* serotype 19A strain in a randomized, blinded, and controlled animal study. As PM affects in particular children under the age of 5 years, infant rats are especially

well suited to mimic pediatric bacterial meningitis, since they reflect the developmental stage of this specific population.

## Material and methods

### Infecting bacterial strain

A ceftriaxone-resistant (in accordance with CLSI guidelines, MIC $\geq$ 2 μg/ml as determined by E-test, [Biomérieux, Nürtingen, Germany]) clinical isolate of *S. pneumoniae*, serotype 19A (identified as SPn28) was used. Bacteria were routinely grown overnight at 37°C in brain heart infusion broth (BHI, BD, USA), adjusted to an optical density ($OD_{570}$) of 0.1, and then further diluted tenfold in fresh and pre-warmed BHI medium. The subculture was grown for 2 h to reach logarithmic growth phase. Colony forming units (CFU) were determined on Columbia sheep blood agar (CSBA) plates.

### Expression and purification of PlyAZ3a[T]

Endolysin was produced as previously described, with minor modification [29, 30]. Briefly, the gene coding for PlyAZ3a[T] was amplified from the genome of *Streptococcus tigurinus* strain AZ3a[T] [31] by PCR using specific primers (5′- ATGAAGAAAAACGACTTATTCATCGACG- 3′ and 5′-ATTTAGTGGTAACAATTAGTCCATCAGGTAATACATC-3'), cloned into plasmid pIN-IIIA and transformed into *E. coli* DH5α. Expression was induced by addition of 2% (w/v) lactose. PlyAZ3a[T] binds strongly to diethylaminoethyl (DEAE). Therefore, crude extracts of PlyAZ3a[T] were purified by fast protein liquid chromatography (FPLC) with HiTrap DEAE FF columns (GE Healthcare Bio-Science AB, Uppsala, Sweden) followed by endotoxin removal to < 1 endotoxin units (EU)/mg (ToxinEraser™ Endotoxin Removal Kit, GenScript, Leiden, Netherlands) and dialysis against phosphate-buffer saline (PBS, pH 7.4) (Slide-A-Lyzer™ Dialysis Flasks, ThermoFisher, Reinach, Switzerland). Residual endotoxin concentration was determined by ToxinSensor™ Chromogenic LAL Endotoxin Assay Kit (GenScript, Leiden, Netherlands) and endolysin concentrations using Pierce™ BCA Protein Assay Kits (Thermo-Fisher, Reinach, Switzerland).

### Characterization of PlyAZ3a[T]

To assess similarities of PlyAZ3a[T] (GenBank: EMG32405.1) to the well-characterized pneumococcal endolysin Cpl-1 (GenBank: NP_044837.1) protein alignment was performed using BLASTp [32]. Killing efficacy of PlyAZ3a[T] was assessed on SPn28 and compared to vancomycin. SPn28 was grown to exponential phase and treated with either PlyAZ3a[T] (1 μg/ml, 100 μg/ml or 200 μg/ml), vancomycin (1 μg/ml) or PBS. Bacteria were incubated statically at 37°C with 5% $CO_2$ and CFU were quantified before treatment, and 1, 2, 4 and 6 h after treatment. Prior to quantification on agar plates, 10 μl of PlyAZ3a[T]-treated samples was added to 90 μl of NaCl 0.85% containing 10% w/v choline (Sigma Aldrich, Buchs, Switzerland) in order to block further activity of the enzyme. Six biological replicates were performed for each condition. The effect of temperature and pH on enzymatic activity of PlyAZ3a[T] was evaluated as previously described by Son et al. [33].

### Assessment of toxicity and inflammation in primary astroglial cell culture

The use of animal for cell culture experiments was approved by the animal care and experimentation committee of the canton of Bern, Switzerland (license no. BE 6/20). Primary astroglial cells were isolated and processed from 2-day old infant Wistar rats (central animal facility of the Medical Faculty of the University of Bern, Switzerland) as previously described [34, 35]

with minor adaptations. After 11 days of cultivation, cells were seeded in 24-well plates (~250'000 cells/well) precoated with poly-L-ornithine (0.01 mg/ml in PBS, Buchs, Sigma-Aldrich). Three days later, astroglial cells were exposed for 24 h to (i) sterile PBS (pH 7.4), (ii) 10 μg/ml lipopolysaccharide (LPS) from E. coli 026:B6 (Sigma Aldrich, Switzerland), (iii) 100 μg/ml PlyAZ3aT or (iv) 2% Triton X-100 (Merck, Darmstadt, Germany). Production of $NO_2^-$ was assessed in the supernatant of cell cultures as previously described [34, 35]. Viability of cells was assessed using an XTT assay, according to manufacturer's instruction (Cell Proliferation Kit II, Sigma Aldrich, Buchs, Switzerland). The experiment was performed twice and in biological replicates of 18 for every condition.

Changes in inducible nitric oxide synthase (iNOS) expression were also assessed by immunofluorescence. Cells were fixed with 4% paraformaldehyde (PFA, Merck, Switzerland) diluted in PBS for 10 min at room temperature, followed by three washing cycles with sterile PBS. Permeabilization was done with 0.1% Triton X -100 (TX-100, Merck, Zug, Switzerland) for 5 min, followed by a 1 h blocking with blocking buffer (0.01% TX-100 + 2% bovine serum albumin, dissolved in PBS). Primary antibodies consisted of anti-GFAP mouse antibodies for astrocytes (dilution 1:1000; ThermoFisher, Reinach, Switzerland) and anti-NCOA3 rabbit antibodies for iNOS (dilution 1:500; Amsbio, Abingdon, UK). Secondary antibodies were goat anti-mouse Cy3 for astrocytes (dilution 1:500; ABCAM, Switzerland) and donkey anti-mouse Cy3 for iNOS (dilution 1:500; ThermoFisher, Switzerland), dissolved in blocking buffer. Secondary antibodies were added after overnight incubation of the primary antibodies at 4°C and three washing cycles with PBS and incubated for additional 2 h at room temperature. 4′,6′-diami-dino-2-phenylindole (DAPI, ThermoFisher, Reinach, Switzerland) was used to visualize cell nuclei. Images were generated using fluorescence microscopy (OLYMPUS CKX53, Olympus LS, Germany).

## Infant rat model of pneumococcal meningitis

All animal experiments were approved by the animal care and experimentation committee of the canton of Bern, Switzerland (license no. BE 5/20). Animals had access to water and food ad libitum and were kept in rooms with controlled temperature (22 ± 2°C) and natural light.

A well-established infant rat model for PM was used as previously described [34–38] and adapted to use the serotype 19A clinical isolate SPn28. SPn28 was passaged 3 times in infant rats before further investigations were performed.

Mixed-sex Wistar rats (outbred, 11-day old) and their dams were purchased from Charles River Laboratories (Sulzfeld, Germany). Bacteria in logarithmic growth phase were used as inoculum and were washed twice by centrifugation (3100xg, 10 min at 4°C) and resuspended in ice cold sterile saline (NaCl 0.85%), followed by dilution to reach the desired $OD_{570}$ of 0.15.

Bacterial inoculation (10 μl containing 3.9 ± 0.25 x $10^7$ CFU/ml SPn28) was performed by injection into the cisterna magna. This amount of inoculum was chosen according to the results of $LD_{90}$ experiments (S1 Fig). CSF sampling was performed by puncture of the cisterna magna using a 30-gauge needle and blood sampling by puncturing the facial vein with a 20-gauge needle and collection of 1 drop (approximately 50 μl). Clinical signs of meningitis arose at 17 hours post infection (hpi), and this correlated with quantitative analysis of bacterial titers in the CSF. This timepoint was chosen for the first treatment dose. Clinical scoring and weighing were performed at 17, 24 and 42 hpi as described previously [36]. Briefly, clinical score was determined as follows: 5 = healthy with normal behavior; 4 = turns upright within 5 s, weight loss and/or appearance of fur; 3 = turns upright within 30 s; 2 = inability to turn upright; 1 = marked diminished motor activity, coma. Animals were sacrificed upon reaching a score of ≤ 2 to meet ethical requirements. At 42 hpi all remaining animals were sacrificed by

intraperitoneal (i.p.) injection of pentobarbital (Esconarkon®, 150 mg/kg, Streuli Pharma AG, Switzerland) followed by perfusion with 4% PFA. Brains were collected for histomorphometric analysis of cortical damage and hippocampal apoptosis, which was performed as previously described [35, 39, 40].

**Pharmacokinetic study.**   In order to assess pharmacokinetic properties of PlyAZ3a[T], infected animals were treated with a single i.p. injection of either 100 mg/kg or 400 mg/kg b.w. PlyAZ3a[T] at 17 hpi. Animals were separated in 2 groups (n = 16 per group) with different sample withdrawal times, in order to avoid excessive sampling frequency from the same animals. CSF and blood samples were taken at 15 min, 4 h and 12 h for the first group and at 2 h, 6 h and 12 h for the second group. If animals reached a clinical score ≤ 2, samples were taken after sacrifice and included into the analysis. Blood was collected in serum Microvettes® 200 Z (Sarstedt, Nümbrecht, Germany).

**Treatment study.**   The procedure for the randomized, blinded and controlled study of PM was as follows. CSF sampling was performed at 17 hpi to confirm meningitis and all animals were randomly allocated into 3 groups using GraphPad (https://www.graphpad.com/quickcalcs/randomize1.cfm). Treatment groups were (i) placebo (sterile PBS), (ii) PlyAZ3a[T] 400 mg/kg b.w. and (iii) Vancomycin 50 mg/kg b.w. (Vancocin, in saline, Teva, Petach Tikwa, Israel). Treatments were applied twice daily (12-hour time interval) through an i.p. injection. All persons involved in animal treatment were blinded regarding treatment groups until the end of the experiment. The primary endpoint was survival at 42 hpi. Secondary endpoints were (i) CFU in CSF and blood at 6 and 25 h post treatment and (ii) extent of cortical necrosis and hippocampal apoptosis in histopathological analysis.

The volume of CSF recovered from each animal during sampling was restricted and ranged from 5 to 15 μl. This resulted into a limit of detection for CFU quantification of $1 \times 10^3$ CFU/ml.

Animals displaying a clinical score of ≤ 2 prior to the time of randomization were sacrificed immediately and excluded from the analysis. Similarly, animals without established meningitis at randomization, defined by bacterial titers in CSF $<1 \times 10^4$ CFU/ml at 17 hpi, were excluded from the analysis.

## Quantification of PlyAZ3a[T] in CSF and blood samples

The amount of PlyAZ3a[T] in CSF and serum samples was assessed by quantitative near infrared Western Blot (NIR-WB). Proteins in CSF or serum were separated using 10% SDS-polyacrylamide gels. Western blotting and immunostaining were performed with WesternBrightTM MCF and MCF-IR kit (Advansta, San Jose, USA) according to manufacturer's instructions. PlyAZ3a[T] was detected as a 41 kD band, by using a primary antibody (custom-produced PlyAZ3a[T]-specific polyclonal rabbit antibody, David's Biotechnology, Regensburg, Germany). Fluorescence was measured using a Fusion FX6 EDGE System (Vilber Lourmat, Collégien, France) with an exposure time of 2 min. Quantification of PlyAZ3a[T] was performed by comparison to a standard curve of PlyAZ3a[T] ranging from 2–0.02 μg using FIJI (version 1.8). The limit of detection was 10 μg/ml.

## Analysis of cytokines in cell culture supernatant

Inflammatory cytokines known to be upregulated in PM (IL-1β, IL-6 and TNF-α) were measured in cell supernatants and CSF samples using a magnetic multiplex assay (Rat Magnetic Luminex® Assay, R&D Systems, Bio-Techne) on a Bio-Plex 200 station (Bio-Rad Laboratory) as reported previously [35, 41].

## Plate lysis assay for endolysin resistance testing

Plate lysis assay was performed as described previously [42–44] to assess resistance and activity against various pneumococcal serotypes. Bacteria recovered from animals was grown to exponential phase and plated on a CSBA to produce uniform bacterial lawns. PlyAZ3a$^T$ (0.5 µg) was spotted onto the plate and incubated overnight. Bacteria were classified as susceptible to PlyAZ3a$^T$ if a clear lysis zone was observed.

## Statistical analysis

Statistical analyses were performed using GraphPad Prism$^®$ (San Diego, USA). Differences in survival were calculated using log-rank tests. Differences between means of non-normally distributed groups (cytokines, cortical necrosis, hippocampal apoptosis) were assessed using Kruskal-Wallis test with Dunn's multiple comparison. Differences in CFU reduction in CSF and blood of animals were calculated using Fisher's exact test. A p value $< 0.05$ was considered statistically significant.

## Results

### PlyAZ3a$^T$ lyses ceftriaxone-resistant *S. pneumoniae in vitro*

PlyAZ3a$^T$ is a bacteriophage-derived endolysin found within the genome of *S. tigurinus* strain AZ3a$^T$ (Accession number GCA_000344275.1). Alignment of the amino acid sequence of PlyAZ3a$^T$ and Cpl-1 (up to now the most extensively characterized pneumococcal endolysin) displayed 80% similarity for the putative catalytic domain and 47% similarity for the putative cell-wall binding domain (Fig 1A), suggesting PlyAZ3a$^T$ might be active against *S. pneumoniae*. Indeed, PlyAZ3a$^T$ exhibit a broad-spectrum lytic activity against *S. pneumoniae*, lysing 18 distinct serotypes including both PCV 13 (3, 5, 6A, 6B, 7F, 9V, 14, 18C, 19A, 19F and 23F) and non-PCV 13 serotypes (11A, 15A, 17F, 20, 22F, 23B and 35B), as tested by plate lysis assay. No resistance to the endolysin activity could be detected among the tested isolates (n = 18). PlyAZ3a$^T$ lysed *S. pneumoniae* efficiently ($>$50% of maximal efficacy) at temperatures ranging from 25–42˚C (Fig 1B) and for pH ranging from 5–7. Optimal activity occurred at pH 6, while pH $\geq$ 9 completely inactivated the enzyme (Fig 1B). In time-kill curve assays, PlyAZ3a$^T$ showed a dose-dependent killing of the ceftriaxone-resistant strain SPn28. The bacteria were not eradicated even after 6 h of exposure to the lysin at the highest concentration of 200 µg/ml (Fig 1C). In contrast, vancomycin displayed a bactericidal effect and reduced counts below the detection limit within the same time period (Fig 1C).

### PlyAZ3a$^T$ failed to control ceftriaxone-resistant pneumococcal meningitis

In primary astroglial cell cultures, no direct toxicity of PlyAZ3a$^T$ was observed (S2A and S2B Fig). Exposure led to a minimal inflammatory response when compared to the inflammatory control LPS as inferred by $NO_2^-$ and cytokine levels (S2C and S2D Fig), suggesting the protein was safe for animal administration. Overall, 38 infant Wistar rats were included in the treatment study, which was performed in 2 independent experiments. At the time of randomization (17 hpi), 8 animals had to be excluded due to a clinical score $\leq$ 2. The 30 remaining animals were randomly allocated to receive either (i) placebo (n = 8), (ii) PlyAZ3a$^T$ (n = 11) or (iii) vancomycin (n = 11) (S3 Fig). Post hoc additional 4 animals not having established meningitis ($<$1 x 10$^4$ CFU/ml in the CSF at 17 hpi) had to be excluded (n = 3 for PlyAZ3a$^T$, n = 1 for vancomycin).

PlyAZ3a$^T$ was not superior to placebo in improving survival of animals with PM (p $>$ 0.05, Fig 2A). In contrast, the standard-of-care antibiotic vancomycin significantly improved

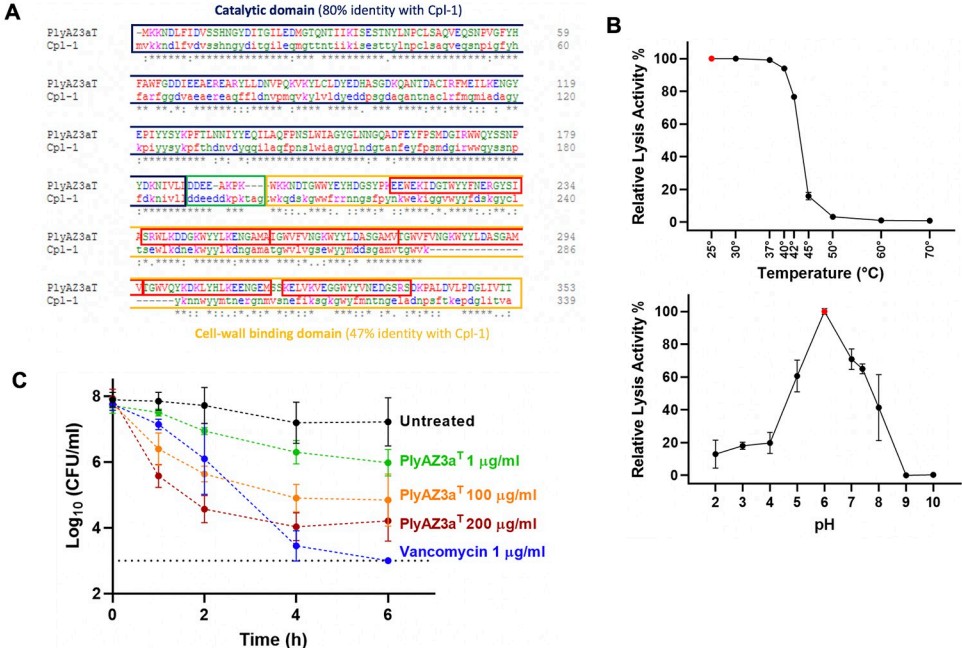

**Fig 1.** **(A)** Alignment of the amino acid sequences of PlyAZ3a$^T$ and Cpl-1. Each domain is highlighted in a different color. The catalytic domain in blue, the cell-wall binding domain in orange and the linker in green. PlyAZ3a$^T$ displays six choline binding repeats marked in red. **(B)** Lytic activity of PlyAZ3a$^T$ after exposure for 10 min to different temperatures and pH. Relative lytic activity was assessed by measuring the reduction in OD$_{570}$ and comparing it to the condition with the maximal reduction (data point marked in red and defined as 100%). **(C)** Time-kill curves for S. pneumoniae strain SPn28 exposed to PlyAZ3a$^T$ or vancomycin. The dotted line represents the limit of detection. CFU, colony forming units. All experiments were performed in triplicate. Data are presented as mean ± standard deviation.

survival with a rate of 70% survival (p = 0.02 versus PBS and PlyAZ3a$^T$, Fig 2A). PlyAZ3a$^T$ did not significantly reduce bacterial loads in the CSF and blood over the course of infection when compared to placebo (p > 0.05, Table 1). On the opposite, vancomycin-treated animals displayed a consistent reduction of bacterial loads in both CSF and blood within the first 6 h after start of therapy (CSF p = 0.0182 and blood p = 0.021 compared to placebo, Table 1). In surviving animals under vancomycin treatment, the infection was cleared within 25 h of treatment (S4 Fig). The kinetics of bacteria in blood of PlyAZ3a$^T$ and placebo treated animals are displayed in S4 Fig. Finally, the extent of cortical necrosis and the number of apoptotic cells of the subgranular zone of the dentate gyrus were similar among all groups (p > 0.05 for all comparisons) (S5A and S5B Fig).

## Reduced bioavailability of PlyAZ3a$^T$ in CSF

We further investigated whether the lack of efficacy of PlyAZ3a$^T$ in the experimental meningitis model was associated either with the emergence of endolysin resistance or with an altered pharmacokinetic profile. First, bacteria recovered from PlyAZ3a$^T$ treated animals (60 colonies in total) were tested for resistance and all remained susceptible to lysis. Second, PlyAZ3a$^T$ was quantified from the serum and CSF of infected animals receiving a single dose of either 100 mg/kg or 400 mg/kg b.w. PlyAZ3a$^T$. PlyAZ3a$^T$ was detected in the serum (Fig 2B), peaked after 2 h of intraperitoneal administration (268 µg/ml for 100 mg/kg group and 785 µg/ml for 400 mg/kg group), and was detected up to 12 h in the 400 mg/kg BW group, albeit at low levels (Fig 2C). In contrast, PlyAZ3a$^T$ was not detected in the CSF of any of the animals tested, at any time point (Fig 2B, limit of detection 10 µg/ml).

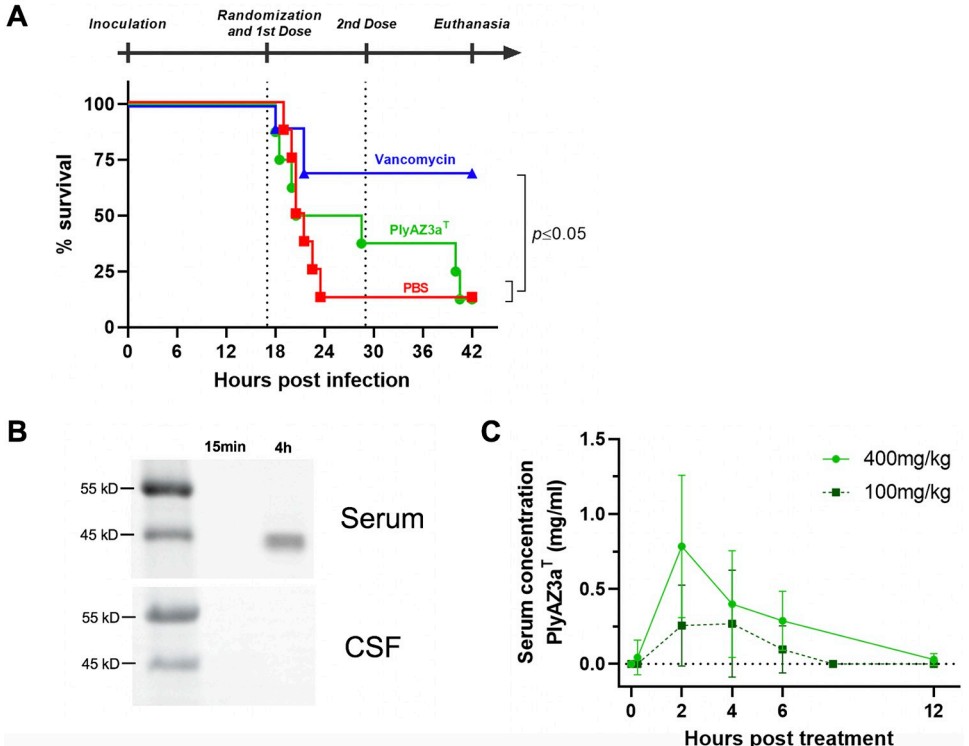

**Fig 2. Treatment of ceftriaxone-resistant pneumococcal meningitis using PlyAZ3a$^T$.** **(A)** Kaplan-Meier survival curve of Wistar rats treated with either PBS (n = 8), 400 mg/kg PlyAZ3a$^T$ (n = 8) or 50 mg/kg Vancomycin (n = 10). Treatment was applied intraperitoneally twice daily. Significance was determined by log-rank test. **(B)** Anti-PlyAZ3a$^T$ near-infrared western blots of serum or CSF samples recovered from one animal treated with a single intraperitoneal injection at 15 min and 4 h after injection. **(C)** Pharmacokinetic assessment of PlyAZ3a$^T$ in serum of Wistar pups (n = 16 per group) after one single intraperitoneal injection. Limit of detection 10 µg/ml. Data are presented as mean ± standard deviation.

## Discussion

Despite the introduction of the pneumococcal conjugate vaccine 13 (PCV13), *S. pneumoniae* serotype 19A remains a major causative of invasive pneumococcal disease and meningitis [22, 45]. In addition, serotype 19A is among the major serotypes associated with penicillin or

**Table 1. Comparison of the decrease of CFU in CSF and blood between treatment groups within the first 6 h after treatment.** (A) Comparison of CFU reduction in CSF and **(B)** in blood. Statistical significance was determined using Fisher's exact test. Animals displaying bacterial loads below the limit of detection throughout the experiment or missing follow-up sampling were excluded from the analysis.

**A) Numbers of animals displaying decrease of CFU in CSF**

| CSF | Decrease | No Decrease | Comparison to PBS |
|---|---|---|---|
| PBS | 1 | 3 | |
| PlyAZ3a$^T$ | 2 | 4 | p > 0.05 |
| Vancomycin | 8 | 0 | p = 0.0182 |

**B) Numbers of animals displaying decrease of CFU in blood**

| CSF | Decrease | No Decrease | Comparison to PBS |
|---|---|---|---|
| PBS | 2 | 5 | |
| PlyAZ3a$^T$ | 6 | 1 | p > 0.05 |
| Vancomycin | 6 | 0 | p = 0.021 |

cephalosporin resistance [22]. Having a wider range of treatment alternatives to combat ceftriaxone-resistant pneumococci is of necessity. Herein, we investigated whether a novel phage lysine recently isolated from *S. tigurinus* strain AZ3a[T] [31] would be effective for the treatment of *S. pneumoniae* serotype 19A either *in vitro* or *in vivo*, in an experimental bacterial meningitis model in infant rats. *In vitro*, PlyAZ3a[T] showed a broad host-range and was able to lyse all tested *S. pneumoniae* serotypes independently from their antibiotic resistance profile. PlyAZ3a[T] induced a quick and dose-dependent reduction of bacterial loads of the ceftriaxone-resistant serotype 19A. Furthermore, the endolysin was safe and did not induce any evident direct toxic effect in astroglial cell cultures. Only minor inflammatory reactions were observed, mostly related to residual endotoxin activity.

*In vivo*, PlyAZ3a[T] failed to improve survival, to reduce bacterial numbers in the CSF or to prevent systemic spreading of the infection in blood compared to placebo. On the other hand, vancomycin significantly increased survival to 70%, which is in accordance with survival rates observed in previous studies when using standard of care treatment in this animal model [34, 35, 38]. PlyAZ3a[T] showed an important pharmacokinetic limitation, as endolysin could not be detected in CSF of endolysin treated animals most likely as a result of its inability to cross the BBB or the low stability of the protein in this environment. A possible solution to bypass this limitation is the use of direct intracisternal application, which is not devoid of problems either, as shown in a previous investigation assessing the pneumococcal endolysin Cpl-1 in the similar experimental setting [28]. There, Cpl-1 exhibited a short half-life time of ~16 min after direct intracisternal application, leading to bacterial regrowth within 4 h and prompting the necessity of repeated intracisternal injections, which on their part where harmful [28]. An intracisternal-sustained release might help to optimize the pharmacokinetic profile. In bacterial meningitis, such an opportunity might be found when an external ventricular drain (EVD) is available, allowing for a continuous delivery of endolysin directly to the CNS thereby circumventing the BBB. Alternatively, innovative galenic formulations (e.g. using liposomes) might improve the ability of the endolysin to cross the BBB allowing for an intravenous administration.

Taken together, our results highlighted the pitfalls and drawbacks of the application of PlyAZ3a[T], and possibly endolysins in general, in bacterial meningitis. Despite a well described break down of the BBB in bacterial meningitis the BBB seems to remain largely impermeable to such phage proteins. As such, endolysins might be more suitable for the treatment of other bacterial infections as it was shown for Cpl-1, such as bacteremia [4, 6], pneumonia [25] or endocarditis [26] where the endolysin can efficiently and readily reach the infection site.

## Supporting information

**S1 Fig. Adaptation of a ceftriaxone-resistant pneumococcal strain (serotype 19A) to the infant rat model of meningitis.** Kaplan-Meier survival curves for $LD_{90}$ finding of the new strain **(A)** before and **(B)** after 3 passages in infant rats. 10 μl of the corresponding inoculum were injected.
(TIF)

**S2 Fig. PlyAZ3a[T] induces a minimal inflammatory reaction with no direct toxicity in confluent primary astroglial cell cultures.** Cells were exposed for 24 h to either PBS (negative control), LPS (positive control), PlyAZ3a[T] or TritonX (membrane disrupting agent). **(A)** Immunohistological staining in astroglial cell cultures, iNOS is stained in green (anti-iNOS), astroglia is stained in red (anti-GFAP) and cell nuclei in blue (4′,6-Diamidin-2-phenylindole, DAPI). **(B)** Viability of cells was confirmed by using an XTT-assay (the mean of PBS treated cells is defined as 100%). **(C)** Production of $NO_2^-$ measured as an index for nitric oxide (NO) release. **(D)** Levels of IL-1β, TNF-α and IL-6. Data are presented as mean ± standard deviation.

Statistical differences between groups were assessed using Kruskal-Wallis test with Dunn's multiple comparisons; *p<0.05, **** p<0.0001.
(TIF)

**S3 Fig. Flowchart of the randomized, blinded and controlled experimental study.**
(TIF)

**S4 Fig. Tracked course of bacterial loads in CSF (black) and blood (purple) in each single animal.** Tracking of bacterial loads displayed for the first six hours or until the end of the experiment (25 hours post treatment). Dots connected with lines represent repeated sampling in one single animal. CFU, colony forming unit.
(TIF)

**S5 Fig. Histopathological assessment of cerebral complications after acute pneumococcal meningitis. (A)** Comparison in percentage of necrotizing cortex between groups and **(B)** number of apoptotic cells in the hippocampus. Animals reaching the end of the 42 hours trial are represented by closed circles and succumbed prematurely to the infection by crosses. Statistical significance was assessed using Kruskal-Wallis test with Dunn's multiple comparisons, ns, not significant.
(TIF)

**S1 Raw images. Raw images of the western blots displayed in Fig 2B.** The red rectangle corresponds to the section displayed in Fig 2B. Lanes not included in the final figure are marked with an "X".
(PDF)

**S1 Table. Overview of the raw data of CFU detected in CSF.** Bacterial loads are displayed as $Log_{10}$ (CFU). Hpt, hours post treatment. N/A, not available due to failed sampling. Limit of detection is 3 $Log_{10}$.
(TIFF)

**S2 Table. Overview of the raw data of CFU detected in blood.** Bacterial loads are displayed as $Log_{10}$ (CFU). Hpt, hours post treatment. N/A, not available due to failed sampling. Limit of detection is 3 $Log_{10}$.
(TIFF)

**S3 Table. Spreadsheet containing raw data to Figs 1 and 2.**
(XLSX)

**S4 Table. Spreadsheet containing raw data to supplemental figures.**
(XLSX)

## Acknowledgments

We acknowledge the outstanding technical assistance of Maria Erhardt, Sandra Nansoz, Franziska Simon, Romano Josi and Jonathan Save. We thank Carlo Casanova and Markus Hilty from the Swiss National Center for Pneumococci (https://www.ifik.unibe.ch/services/pneumococcal_center/index_eng.html) for generously providing us a ceftriaxone-resistant *S. pneumoniae* (serotype 19A). We thank Andrea Zbinden for having kindly provided us with the *Streptococcus tigurinus* strain AZ3a[T].

## Author Contributions

**Conceptualization:** Luca G. Valente, Denis Grandgirard, Stephan M. Jakob, David R. Cameron, Yok-Ai Que, Stephen L. Leib.

**Formal analysis:** Luca G. Valente.

**Funding acquisition:** Yok-Ai Que, Stephen L. Leib.

**Investigation:** Luca G. Valente, Ngoc Dung Le, Melissa Pitton, Gabriele Chiffi, Denis Grandgirard, Grégory Resch.

**Methodology:** Luca G. Valente, Ngoc Dung Le, David R. Cameron.

**Project administration:** Luca G. Valente.

**Supervision:** Yok-Ai Que, Stephen L. Leib.

**Writing – original draft:** Luca G. Valente.

**Writing – review & editing:** Luca G. Valente, Ngoc Dung Le, Melissa Pitton, Gabriele Chiffi, Denis Grandgirard, Stephan M. Jakob, David R. Cameron, Grégory Resch, Yok-Ai Que, Stephen L. Leib.

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
