## [Decision Letter · Decision Letter 0]

16 Feb 2022

PONE-D-22-01149Efficacy assessment of a novel endolysin PlyAZ3aT for the treatment of ceftriaxone-resistant pneumococcal meningitis in an infant rat modelPLOS ONE

Dear Dr. Leib,

Thank you for submitting your manuscript to PLOS ONE. After careful consideration, we feel that it has merit but does not fully meet PLOS ONE’s publication criteria as it currently stands. Therefore, we invite you to submit a revised version of the manuscript that addresses the points raised during the review process.

We look forward to receiving your revised manuscript.

Kind regards,

Rosa del Campo

Academic Editor

PLOS ONE

Journal Requirements:

Reviewers' comments:

Reviewer's Responses to Questions

**Comments to the Author**

1. Is the manuscript technically sound, and do the data support the conclusions?

Reviewer #1: Yes

Reviewer #2: Yes

Reviewer #3: No

2. Has the statistical analysis been performed appropriately and rigorously? 

Reviewer #1: Yes

Reviewer #2: Yes

Reviewer #3: Yes

3. Have the authors made all data underlying the findings in their manuscript fully available?

Reviewer #1: Yes

Reviewer #2: Yes

Reviewer #3: Yes

4. Is the manuscript presented in an intelligible fashion and written in standard English?

Reviewer #1: Yes

Reviewer #2: Yes

Reviewer #3: No

5. Review Comments to the Author

Reviewer #1: In this work, the efficacy of the recombinant endolysin PlyAZ3aT identified in the genome of Streptococcus tigurinus AZ3aT was evaluated in an infant rat model of pneumococcal meningitis. In general, this is a well-organized manuscript whose scope is worthy of investigation for this journal. However, the results showed that the endolysin PlyAZ3aT did not improve the survival of infant rats when compared with animals treated with vancomycin. Although the use of endolysins has been reported as a promising solution against antibiotic resistant pathogenic bacteria, the results obtained in this work suggest that the endolysin PlyAZ3aT is not a real alternative to treat meningitis provoked by Streptococcus pneumoniae serotype 19A. In fact, the results showed that the use of the antibiotic vancomycin is a better alternative to treat the infection produced by this ceftriaxon-resistant pneumococcal strain in rats.

The main concern of this study is that PlyAZ3aT did not exert a bactericidal effect in vitro as the growth of S. pneumoniae was not inhibited after 6 h of exposure using the highest concentration in antibactericidal plate assays (200 ug/ml). Consequently, this fact hinders the possible antimicrobial effect of this endolysin in rats inoculated with S. pneumoniae. The second issue is the failure of this endolysin to cross the blood brain barrier as it could not be detected in cerebrospinal fluid. In the discussion section, it was proposed the use of liposomes for endolysin delivery across the blood brain barrier in order to improve its efficacy and enhance its action in meningitis treatment. Therefore, the manner in which this work is presented is not enough to have an own identity as a primary research article. In my opinion, a new assay using liposomes or other mechanisms as endolysin delivery vehicles should be tested in an infant rat model of pneumococcal meningitis to elucidate whether PlyAZ3aT is active in vivo against S. pneumoniae. Otherwise, this manuscript should be considered as a letter to editor to explain the ineffectiveness of this endolysin and avoid developing uselessness assays with this endolysin against S. pneumoniae. The third problem is related to the inflammatory response observed in supernatants from primary astroglial cell culture treated with a low dose of PlyAZ3aT (100 ug/ml). In this sense, levels of cytokines (IL-1β, IL-6 and TNF-α) were statistically significant when compared with those obtained in control cultures treated with PBS and it was attributed to residual endotoxin activity. Finally, it should be explained why the detection limit of bacterial counts is 3 log10 (CFU/ml).

There are other specific comments:

Abstract

- Line 30. The following sentence seems a headline: "Randomized, blinded and controlled experimental study in infant Wistar rats".

Results

- Line 240. It should be specified that PlyAZ3aT is a bacteriophage-derived endolysin found in the genome of Streptococcus tigurinus.

Discussion

- The discussion should include more studies regarding the application of endolysin in animal models.

Tables

- The comparison of the decrease of CFU in CSF and blood shown in Table 1 is not representative. However, data from Table S1 should be plotted in a graph. The abbreviation N/A should be explained.

Figures

- Figure 1. The alignment of amino acid sequences of the catalytic and cell-wall binding domains of endolysins PlyAZ3aT and Cp-1 should be included in Fig.1A instead of the schematic structure of PlyAZ3aT in order to support the information explained in lines 240-244 (Results).

Reviewer #2: The manuscript submitted by Valente and colleagues describes the use of the streptococcal endolysin PlyAZ3a to treat pneumococcal meningitis in an animal model. Disease was induced in animals by inoculation of Streptococcus pneumoniae and 17 h later animals were treated with endolysin. Survival of treated animals was not higher than those treated with PBS (negative control). Additionally, it was shown that endolysin was not found in the cerebrospinal fluid, which would explain the failure in the animals’ recovery.

General comments: the manuscript is well written and organized. The subject of the work is of interest due the huge problem derived from multiresistant bacteria. All experiments have been done with proper controls. Although not positive results were obtained from this work the conclusions are very valuable.

Specific comments:

Lane 115. The antimicrobial activity of endolysin was compared with vancomicyn. Did you compare with endolysin Cpl-1? How different is the activity of PlyAZ3a and Cpl-1? It might explain the different activity in vivo.

Line 279. PlyAZ3 did not reduced the bacteria in CFS. Did you test the activity of endolysin in vitro in a solution similar to CFS? A low activity or stability of the protein in CFS would explain this result. Actually Cpl-1 has a short half-life time.

Line 334. The inability to cross the BBB or the low stability of the protein in this environment.

Specific conditions in CFS also might explain the low activity of the protein but in terms of bacteria metabolic state. Are bacteria growing actively in CFS?

Reviewer #3: The manuscript "Efficacy assessment of a novel endolysin…" by Valente et al, deals with the evaluation of the novel endolysin PlyAZ3aT as a therapeutic agent for treating pneumococcal meningitis in an animal model of infection. The objective is undoubtedly important because of the high incidence of mortality caused by this type of infection, especially in children, and by the increase of antibiotic resistance in certain pathogenic bacterial strains. A key factor adding to the difficulty of the challenge posed is the fact that the niche where the pneumococci causing the infection develop is the cerebrospinal fluid (CSF), which is protected by the blood-brain barrier (BBB).

Faced with this real challenge, the authors have tested the efficacy of the new endolysin in some in vitro experiments, and then in a rat model. But the conclusions of these assays could not be clearer. In their own words:

"PlyAZ3aT was not superior to placebo in improving survival of animals with pneumococcal meningitis“ (line 276).

"PlyAZ3aT did not significantly reduce bacterial loads in the CSF and blood over the course of infection when compared to placebo” (line 278).

“In contrast, PlyAZ3aT was not detected in the CSF in any of the animals tested, at any time point” (line 312).

These negative results do not admit any other interpretation and authors can only speculate on the reasons for these data and discuss possible alternatives to try to make this enzyme effective with another experimental protocol.

Because of these strong negative results, this manuscript does not provide any promising data to meet the proposed objectives. It is evident that it could only be evaluated positively if the authors succeed in developing an alternative method of enzyme delivery that reaches the CSF, and demonstrates that the chosen endolysin has sufficiently potent bactericidal effect to achieve a bacterial lethality that would make the treatment successful.

Although the substance of the evaluation is reflected above, the manuscript contains other serious flaws that deserve some comments:

· Authors claim that the endolysin PlyAZ3aT “was found within the genome of a particular strain of Streptococcus tigurinus”, but it is no comment whether the origin was the proper bacterium or a temperate phage integrated in the chromome.

· According to the alignment between the amino acid sequences of PlyAZ3aT and Cpl-1 (Fig. 1A), both enzymes displayed 47% amino acid similarity for the putative cell wall binding domain (CBD). But the CBD of Cpl-1 is built on six choline binding repeats, which have been demonstrated essential to be fully active for the whole enzyme, as well as in other examples of pneumococcal murein hydrolases, both from bacterial and phage origin. In the case of PlyAZ3aT a single “Cholin_bind_3” (from residue 225 to 296) is depicted in such Fig. 1A. Does it mean that there are no other choline binding repeats in this domain? What are the relevant characteristics of this domain beyond the single “Cholin_bind_3”?

· From the in vitro results presented in this study, it appears evident that PlyAZ3aT is much less active than Cpl-1, in terms of bactericidal activity. The reasons come, most likely, from the comments explained above. Thus, the eventual use of PlyAZ3aT as therapeutic agent against pneumococci does not represent any improvement compared with other endolysins already published.

In summary, the success of the ambitious goal set by the authors would only be achieved with an enzyme (endolysin) very active against pneumococcus and with a formulation capable to cross the BBB and reach the CSF.

6. PLOS authors have the option to publish the peer review history of their article (what does this mean?). If published, this will include your full peer review and any attached files.

Reviewer #1: No

Reviewer #2: No

Reviewer #3: No

---

## [Author Response · Author response to Decision Letter 0]

28 Mar 2022

RE: Response to review: Efficacy assessment of a novel endolysin PlyAZ3aT for the treatment of ceftriaxone-resistant pneumococcal meningitis in an infant rat model’.

Dear Dr. del Campo

We thank the editor and reviewers for their helpful suggestions. Below, we have included a point-by-point response to each comment.

Reviewer #1: 

In this work, the efficacy of the recombinant endolysin PlyAZ3aT identified in the genome of Streptococcus tigurinus AZ3aT was evaluated in an infant rat model of pneumococcal meningitis. In general, this is a well-organized manuscript whose scope is worthy of investigation for this journal. 

Answer: We appreciate the positive response of reviewer #1.

The main concern of this study is that PlyAZ3aT did not exert a bactericidal effect in vitro as the growth of S. pneumoniae was not inhibited after 6 h of exposure using the highest concentration in antibactericidal plate assays (200 ug/ml). Consequently, this fact hinders the possible antimicrobial effect of this endolysin in rats inoculated with S. pneumoniae. 

Answer: Thank you for this pertinent comment. Even if the endolysin is not able to totally eradicate the infection in vitro we hypothesized, that, in vivo, reducing the bacterial loads in the initial phase of the disease would let enough time to the host to mount a sufficiently robust immune answer able to eradicate the remaining living bacteria. However, we do agree to the conclusion that this endolysin (in its current formulation) does not seem suitable in this infection setting and that the results demonstrated in this manuscript are in essence negative results. 

The second issue is the failure of this endolysin to cross the blood brain barrier as it could not be detected in cerebrospinal fluid. In the discussion section, it was proposed the use of liposomes for endolysin delivery across the blood brain barrier in order to improve its efficacy and enhance its action in meningitis treatment. Therefore, the manner in which this work is presented is not enough to have an own identity as a primary research article. In my opinion, a new assay using liposomes or other mechanisms as endolysin delivery vehicles should be tested in an infant rat model of pneumococcal meningitis to elucidate whether PlyAZ3aT is active in vivo against S. pneumoniae. Otherwise, this manuscript should be considered as a letter to editor to explain the ineffectiveness of this endolysin and avoid developing uselessness assays with this endolysin against S. pneumoniae. 

Answer: We indeed suggest another model for delivery across the blood brain barrier, using liposome or direct intracisternal injection. However, in the discussion, we also suggested that, as such, endolysins might be more suitable for the treatment of other bacterial infections such as bacteremia, pneumonia or endocarditis. The actual results we provide do not warrant enough ethical justification for the use of additional animals to test this endolysin in the same setting. Still, in our opinion, our negative results in the in vivo model of meningitis are of importance to the scientific community and this is why we submitted the current manuscript. To our opinion, it is important that negative results obtained from the use of animals are also published.

The third problem is related to the inflammatory response observed in supernatants from primary astroglial cell culture treated with a low dose of PlyAZ3aT (100 ug/ml). In this sense, levels of cytokines (IL-1β, IL-6 and TNF-α) were statistically significant when compared with those obtained in control cultures treated with PBS and it was attributed to residual endotoxin activity. 

Answer: The statement is true; however, we don’t perceive a question in it. To our opinion, the reason for the observed residual inflammatory reaction is the most plausible.

Finally, it should be explained why the detection limit of bacterial counts is 3 log10 (CFU/ml).

Answer: This is due to the small volume of CSF which can be recovered from infant wistar rat, which ranges from 5 to 15 �l per sampling, limiting the amount of sample volume which can be used to quantify the bacterial loads. We routinely dilute 5ul CSF into 495ul PBS and plate out 100ul of the first dilution, this leads to the limit of detection used in this project. 

Measure: We added the following text to better clarify this fact (lines 200 - 202): The volume of CSF recovered from each animal during sampling was restricted and ranged from 5 to 15 µl. This resulted into a limit of detection for CFU quantification of 1 x 103 CFU/ml.

There are other specific comments:

Abstract

- Line 30. The following sentence seems a headline: "Randomized, blinded and controlled experimental study in infant Wistar rats".

Answer: We acknowledge the impression that the sentence seems a headline.

Measure: We changed the sentence to (line 28). Efficacy of PlyAZ3aT was assessed in a randomized, blinded and controlled experimental study in infant Wistar rats.

Results

- Line 240. It should be specified that PlyAZ3aT is a bacteriophage-derived endolysin found in the genome of Streptococcus tigurinus.

Answer: We thank the reviewer for this useful comment.

Measure: The sentence has been changed accordingly (line 242). PlyAZ3aT is a bacteriophage-derived endolysin found within the genome of S. tigurinus strain AZ3aT.

Discussion

- The discussion should include more studies regarding the application of endolysin in animal models.

Answer: We thank the reviewer for this remark.

Measure: An additional sentence and the references have been added (lines 355 and 356). As such, endolysins might be more suitable for the treatment of other bacterial infections as it was shown for Cpl-1, such as bacteremia (4,27), pneumonia (25) or endocarditis (26) where the endolysin can efficiently and readily reach the infection site.

Tables

- The comparison of the decrease of CFU in CSF and blood shown in Table 1 is not representative. However, data from Table S1 should be plotted in a graph. 

Answer: We share the revisers concern and are thankful for the suggestion plot data from Table S1 in a graph. 

Measure: Results of Table S1 and S2 are plotted in a graph shown in Figure S4. It is clearly stated that animals were excluded from this analysis and all the raw data is presented in Table S1 and S2. 

The abbreviation N/A should be explained.

Answer: We concur

Measure: Additional details have been added (lines 569 and 573). Hpt, hours post treatment. N/A, not available due to failed sampling. Limit of detection is 3 Log10.

Figures

- Figure 1. The alignment of amino acid sequences of the catalytic and cell-wall binding domains of endolysins PlyAZ3aT and Cp-1 should be included in Fig.1A instead of the schematic structure of PlyAZ3aT in order to support the information explained in lines 240-244 (Results).

Answer: we thank the reviewer for pointing out this issue in Fig.1A.

Measure: The figure was adapted accordingly, and additional information was added in the figure caption. Alignment of the amino acid sequences of PlyAZ3aT and Cpl-1. Each domain is highlighted in a different color. The catalytic domain in blue, the cell-wall binding domain in orange and the linker in green. PlyAZ3aT displays six choline binding repeats marked in red.

Reviewer #2: 

General comments: the manuscript is well written and organized. The subject of the work is of interest due the huge problem derived from multiresistant bacteria. All experiments have been done with proper controls. Although not positive results were obtained from this work the conclusions are very valuable.

Answer: We appreciate the positive response of reviewer #2.

Specific comments:

Lane 115. The antimicrobial activity of endolysin was compared with vancomicyn. Did you compare with endolysin Cpl-1? How different is the activity of PlyAZ3a and Cpl-1? It might explain the different activity in vivo.

Answer: We agree that this might be an interesting comparison. Unfortunately, we did not have access to Cpl-1 thus this comparison was not done. However, results from the study performed by Grandgirard et al. are hard to compare with this study due to several differences and thus it remains unclear whether there is a real difference between Cpl-1 and PlyAZ3aT. First, a different bacterial strain was used (ceftriaxone-susceptible serotype 3 vs. ceftriaxone-resistant serotype 19A) and PlyAZ3aT was not tested on pneumococcus serotype 3. Second, a different quantification method for the endolysin was used (NIR-WB vs. WB and spot densitometry) displaying different limits of detection. 

Line 279. PlyAZ3 did not reduced the bacteria in CFS. Did you test the activity of endolysin in vitro in a solution similar to CFS? A low activity or stability of the protein in CFS would explain this result. Actually Cpl-1 has a short half-life time.

Answer: In vitro studies with CSF or "CSF-like" media are difficult to perform in static conditions, since the pH of CSF increases significantly after sampling from patients or animals and that these changes in pH after sampling of CSF may affect survival, growth and antibiotic killing of bacterial strains in vitro (DOI: 10.1159/000444263). To tackle this issue, we have performed in vitro tests using a wide range of pHs and temperatures, and found that the activity was preserved in those conditions that can be estimated as similar in the CSF. We agree that, in analogy to Cpl-1, the half-life of PlyAZ3 is probably short. In order to prevent issues with the short half-life time we chose to apply higher amounts of the endolysin by intraperitoneal injections. By choosing this approach we were able to achieve a sustained release lasting for more than six hours as displayed in Figure 2C. 

Line 334. The inability to cross the BBB or the low stability of the protein in this environment.

Answer: We thank the reviewer for this suggestion.

Measure: The sentence has been changed accordingly (line 339). PlyAZ3aT showed an important pharmacokinetic limitation, as endolysin could not be detected in CSF of endolysin treated animals most likely as a result of its inability to cross the BBB or the low stability of the protein in this environment.

Specific conditions in CFS also might explain the low activity of the protein but in terms of bacteria metabolic state. Are bacteria growing actively in CFS?

Answer: We agree, it might well be that specific conditions of the CSF have an impact. However, as can be seen in Figure S4 bacteria do actively grow in CSF and thus it is not likely that they are in a metabolically dormant state.

Reviewer #3: 

The manuscript "Efficacy assessment of a novel endolysin…" by Valente et al, deals with the evaluation of the novel endolysin PlyAZ3aT as a therapeutic agent for treating pneumococcal meningitis in an animal model of infection. The objective is undoubtedly important because of the high incidence of mortality caused by this type of infection, especially in children, and by the increase of antibiotic resistance in certain pathogenic bacterial strains. A key factor adding to the difficulty of the challenge posed is the fact that the niche where the pneumococci causing the infection develop is the cerebrospinal fluid (CSF), which is protected by the blood-brain barrier (BBB).

Faced with this real challenge, the authors have tested the efficacy of the new endolysin in some in vitro experiments, and then in a rat model. But the conclusions of these assays could not be clearer. In their own words:

"PlyAZ3aT was not superior to placebo in improving survival of animals with pneumococcal meningitis “(line 276).

"PlyAZ3aT did not significantly reduce bacterial loads in the CSF and blood over the course of infection when compared to placebo” (line 278).

“In contrast, PlyAZ3aT was not detected in the CSF in any of the animals tested, at any time point” (line 312).

These negative results do not admit any other interpretation and authors can only speculate on the reasons for these data and discuss possible alternatives to try to make this enzyme effective with another experimental protocol.

Because of these strong negative results, this manuscript does not provide any promising data to meet the proposed objectives. It is evident that it could only be evaluated positively if the authors succeed in developing an alternative method of enzyme delivery that reaches the CSF, and demonstrates that the chosen endolysin has sufficiently potent bactericidal effect to achieve a bacterial lethality that would make the treatment successful.

Answer: We agree with the comments. Nevertheless, we think it is of utter importance to publish negative results because it is a crucial step towards limiting publication bias. Especially, considering that animals were used for these experiments, it is ethically important that the scientific community have access the results, even if negative by essence. We had a clear hypothesis and tested it adequately. Our results inform the scientific community that the use of endolysins as therapeutic agents can have some pitfalls which have to be considered, one of them being limited penetration into the CSF.

Although the substance of the evaluation is reflected above, the manuscript contains other serious flaws that deserve some comments:

Authors claim that the endolysin PlyAZ3aT “was found within the genome of a particular strain of Streptococcus tigurinus”, but it is no comment whether the origin was the proper bacterium or a temperate phage integrated in the chromome.

Answer: We agree with the reviewer’s comment

Measure: In line 241 we stated that the endolysin was found within the genome of S. tigurinus AZ3aT. Thus, it is most likely that this endolysin derives from a temperate phage. Additional details have been added (line 242). PlyAZ3aT is a bacteriophage-derived endolysin found within the genome of S. tigurinus strain AZ3aT (Accession number GCA_000344275.1). 

According to the alignment between the amino acid sequences of PlyAZ3aT and Cpl-1 (Fig. 1A), both enzymes displayed 47% amino acid similarity for the putative cell wall binding domain (CBD). But the CBD of Cpl-1 is built on six choline binding repeats, which have been demonstrated essential to be fully active for the whole enzyme, as well as in other examples of pneumococcal murein hydrolases, both from bacterial and phage origin. In the case of PlyAZ3aT a single “Cholin_bind_3” (from residue 225 to 296) is depicted in such Fig. 1A. Does it mean that there are no other choline binding repeats in this domain? What are the relevant characteristics of this domain beyond the single “Cholin_bind_3”?

Answer: Similar to Cpl-1, also PlyAZ3aT displays six choline binding repeats with the consensus sequence TGW-b-(K,Q)DNGSWYYLN-x-SG-z-M-x1-2. 

Measure: This information was also included in Figure 1A. 

From the in vitro results presented in this study, it appears evident that PlyAZ3aT is much less active than Cpl-1, in terms of bactericidal activity. The reasons come, most likely, from the comments explained above. Thus, the eventual use of PlyAZ3aT as therapeutic agent against pneumococci does not represent any improvement compared with other endolysins already published.

Answer: Our data does not allow such a conclusion as we were not able to test Cpl-1 on the given strain (ceftriaxone-resistant serotype 19A) and in the same setting. 

In summary, the success of the ambitious goal set by the authors would only be achieved with an enzyme (endolysin) very active against pneumococcus and with a formulation capable to cross the BBB and reach the CSF.

Answer: We agree to this and our search for better treatment options in bacterial meningitis will continue.

In light of these improvements, we hope that the manuscript is now ready for publication in PLOS ONE.

Sincerely yours, 

Stephen L. Leib

---

## [Editor Report · Decision Letter 1]

30 Mar 2022

Efficacy assessment of a novel endolysin PlyAZ3aT for the treatment of ceftriaxone-resistant pneumococcal meningitis in an infant rat model

PONE-D-22-01149R1

Dear Dr. Leib,

We’re pleased to inform you that your manuscript has been judged scientifically suitable for publication and will be formally accepted for publication once it meets all outstanding technical requirements.

Kind regards,

Rosa del Campo

Academic Editor

PLOS ONE
---

## [Editor Report · Acceptance letter]

4 Apr 2022

PONE-D-22-01149R1 

Efficacy assessment of a novel endolysin PlyAZ3aT for the treatment of ceftriaxone-resistant pneumococcal meningitis in an infant rat model 

Dear Dr. Leib:

I'm pleased to inform you that your manuscript has been deemed suitable for publication in PLOS ONE. Congratulations! Your manuscript is now with our production department. 

Kind regards, 

on behalf of

Dr. Rosa del Campo 

Academic Editor

PLOS ONE